# Dexamethasone in the Treatment of COVID-19: Primus Inter Pares?

**DOI:** 10.3390/jpm11060556

**Published:** 2021-06-15

**Authors:** Vasiliki Romanou, Evangelia Koukaki, Vasiliki Chantziara, Panagiota Stamou, Alexandra Kote, Ioannis Vasileiadis, Antonia Koutsoukou, Nikoletta Rovina

**Affiliations:** 1st Department of Respiratory Medicine, Medical School, National and Kapodistrian University of Athens and “Sotiria” Chest Disease Hospital, 11527 Athens, Greece; vassoromanou@gmail.com (V.R.); e.koukaki@yahoo.gr (E.K.); vchantziara@yahoo.gr (V.C.); giwtastam@live.co.uk (P.S.); akote_1990@hotmail.com (A.K.); ioannisvmed@yahoo.gr (I.V.); koutsoukou@yahoo.gr (A.K.)

**Keywords:** dexamethasone, corticosteroids, COVID-19

## Abstract

Coronavirus disease 2019 (COVID-19), caused by severe acute respiratory syndrome coronavirus 2 (SARS-CoV-2), has rapidly spread globally, becoming a huge public health challenge. Even though the vast majority of patients are asymptomatic, some patients present with pneumonia, acute respiratory distress syndrome (ARDS), septic shock, and death. It has been shown in several studies that the severity and clinical outcomes are related to dysregulated antiviral immunity and enhanced and persistent systemic inflammation. Corticosteroids have been used for the treatment of COVID-19 patients, as they are reported to elicit benefits by reducing lung inflammation and inflammation-induced lung injury. Dexamethasone has gained a major role in the therapeutic algorithm of patients with COVID-19 pneumonia requiring supplemental oxygen or on mechanical ventilation. Its wide anti-inflammatory action seems to form the basis for its beneficial action, taming the overwhelming “cytokine storm”. Amid a plethora of scientific research on therapeutic options for COVID-19, there are still unanswered questions about the right timing, right dosing, and right duration of the corticosteroid treatment. The aim of this review article was to summarize the data on the dexamethasone treatment in COVID-19 and outline the clinical considerations of corticosteroid therapy in these patients.

## 1. Introduction

Coronavirus disease 2019 (COVID-19), caused by severe acute respiratory syndrome coronavirus 2 (SARS-CoV-2), has rapidly spread around the world since its first appearance in China [1,2,3]. Although the vast majority of patients present with mild symptoms, about 15% of COVID-19 cases become severe [4], with approximately 31–41.8% of hospitalized patients rapidly developing acute respiratory distress syndrome (ARDS) [1,2] with a subsequent increased risk of death. The mortality rates referring to critically ill COVID-19 patients range between 26% and 61.5% around the world [1,2,3,4].

Several studies have documented that the severity and clinical outcomes are related to dysregulated antiviral immunity and enhanced and persistent systemic inflammation [5,6,7,8,9], as highlighted by a decreased expression of interferons I/III and hyperinflammatory responses involving the release of proinflammatory cytokines (e.g., IL-6, IL-1, TNF, IL-8, and MCP-1) that are sustaining in time (cytokine storm). Consequently, antiviral and anti-inflammatory therapies have been implemented in the pursuit of timely and effective therapies. As yet, though, no specific pharmacological treatments have been proven efficacious for the treatment of COVID-19 [10].

In the context of inflammation being a key role player in the evolvement of severe disease, corticosteroids have been used for the treatment of COVID-19 patients, as they are reported to elicit benefits by reducing lung inflammation and inflammation-induced lung injury. Early enough, in the first pandemic wave, corticosteroids were introduced in the treatment of critically ill COVID-19 patients with respiratory failure in low-to-moderate doses and for short courses [4,11,12,13]. Wu et al. showed that the corticosteroid treatment in patients with ARDS reduced the mortality risk (46% in patients receiving corticosteroids vs. 61.8% in patients not receiving corticosteroids [14]. Furthermore, the Surviving Sepsis Campaign suggests the use of low-dose corticosteroid therapy in COVID-19 patients with refractory shock to suppress the cytokine storm caused by SARS-CoV-2 and to improve peripheral vasodilation [15]. With the RECOVERY trial, a significant benefit of glucocorticoid dexamethasone was shown in patients with COVID-19 on mechanical ventilation or oxygen alone at the time of randomization [16]. Based on the positive results of the RECOVERY trial, the National Institutes of Health (NIH) COVID-19 Treatment Guidelines and the World Health Organization (WHO) approved the findings on the beneficial use of dexamethasone in treating critically ill patients with COVID-19 and recommended the dexamethasone treatment in hospitalized patients with respiratory failure requiring a noninvasive/high-flow oxygen device or invasive ventilation [17]. Nonetheless, several studies have reported deleterious effects caused by the corticosteroid treatment that was associated with the use of an increased dose of corticosteroids [18,19,20], the delay of SARS-CoV-2 viral clearance in patients receiving corticosteroids [18,21], and the increased risk of secondary infections and complications due to the corticosteroid treatment. Therefore, the clinical benefit of this treatment in COVID-19 patients still remains controversial.

## 2. Current Evidence

Beyond doubt, the RECOVERY trial showed a clear benefit of dexamethasone in critically ill patients with COVID-19 on mechanical ventilation at the time of randomization as compared to the usual care (28-day mortality of 29.3% vs. 41.4%). Similarly, patients on oxygen alone in the dexamethasone group, when compared to the usual care group, had a lower risk of progression to invasive mechanical ventilation (risk ratio, 0.79; 95% CI, 0.64 to 0.97), a shorter duration of hospitalization (median, 12 days vs. 13 days), and a greater probability of being alive at discharge within 28 days (rate ratio, 1.10; 95% CI, 1.03 to 1.17). No benefit was shown in patients with a milder disease without respiratory failure (28-day mortality of 17.8% vs. 14%) [16].

Two more observational trials of corticosteroids for COVID-19 [14,22] and four review and meta-analysis studies [23,24,25,26] showed evidence of benefits in mortality. Of note, Pasin et al. in their metanalysis [24] of the five basic studies on corticosteroids ikn COVID-19 [16,27,28,29,30] coincided with the RECOVERY trial results in that the survival benefits of corticosteroids exist for mechanically ventilated patients and not for patients with no oxygen need. In line with these positive results, subgroups of COVID-19 patients with ARDS [31,32], increased inflammatory markers [33], or over 60 years of age [30] had the highest benefits from corticosteroids. Improved clinical symptoms, radiological findings, and the respiratory failure of patients with COVID-19 were also reported [34,35]. In a small randomized controlled trial, improved clinical outcomes were reported in patients with COVID-19 receiving a short course of methylprednisolone [13]. Wang et al. [35] also found that the use of methylprednisolone (1 to 2 mg/kg/day for 5–7 days) in twenty-six severe COVID-19 patients was associated with a reduced length of hospitalization and ICU stay.

Collectively, a prospective meta-analysis of the trials by the WHO, with a total of 1703 patients (59% from the RECOVERY trial), confirmed a reduction in the 28-day mortality (summary odds ratio (OR) 0.66, 95% CI 0.53–0.82; *p* < 0.001) [23].

Nevertheless, the mortality benefits of corticosteroids were not confirmed in a series of other studies [27,28,30,36,37,38,39]. The retrospective cohorts of COVID-19 patients, mostly from the first wave in China [18,19,40,41], reported an increased mortality in severely ill patients taking high-dose corticosteroids. Lu, X. et al. [19] mentioned that a 10-mg increase in hydrocortisone resulted in a 4% increase in the risk of death. In accordance were the results of one meta-analysis study [20] based on low-quality evidence with a high variability. Table 1, Table 2, Table 3 and Table 4 summarizes the trials on corticosteroid treatments in COVID-19 patients.

These results should be interpreted with caution owing to the potential bias and confounding factors, since they mostly derive from small-sized single-center observational studies; they include patients of different disease stages or severities, and furthermore, they refer to corticosteroid treatments other than dexamethasone. It is noteworthy that dexamethasone showed favorable results in the RECOVERY trial, especially in the critically ill patients, setting a novel positive view on corticosteroid use in respiratory viral infections in these patients. Hence, understanding dexamethasone’s mode of action merits the uppermost importance.

## 3. Dexamethasone in COVID-19: Mode of Action

Corticosteroids are known to temper down the host inflammatory response in the lungs, which may lead to acute lung injury and ARDS [52]. They exert their anti-inflammatory effects by binding to their receptor, the glucocorticoid receptor-GR. GR is a ligand-activated transcription factor whose signaling plays a key role in the modulation of numerous biological functions of the immune cells. Upon ligand binding, GR translocates to the nucleus, where it can enhance transcription by binding to glucocorticoid response elements (GREs). On the other hand, GR, when tethering directly to negative glucocorticoid response elements (nGREs) or GRE half-sites or by binding to transcription factors such as NF-κB, may repress the expression of several genes [53]. Thus, it is evident that the corticosteroid treatment is a double-edged sword, which may induce or inhibit inflammatory cell activity. Several studies have shown that corticosteroids may inhibit T-lymphocyte immunity, an important antiviral mechanism, resulting in persistent viral replication and a delay in viral clearance. At the same time, the suppression of proinflammatory cytokines such as TNF, interleukins, and other genes, such as cyclooxygenase-2 and inducible nitric oxide synthase (iNOS), and the inhibition of the neutrophils, macrophages, and lymphocytes inflammatory activity are documented as effects of corticosteroid use [54].

Dexamethasone has both anti-inflammatory and immunosuppressive effects. Specifically, when binding to GR, it initiates a cascade of immune cell responses that lead to the suppression of proinflammatory cytokines, such as IL-1, IL-6, IL-8, TNF, and IFN-γ, by inhibiting gene transcription [54]. In the case of SARS-CoV-2 infection, those are the produced proinflammatory cytokines that result in local tissue inflammation and in the so-called “cytokine storm” [55]. Furthermore, dexamethasone exerts anti-inflammatory effects by inhibiting macrophage activation [56], neutrophil’s adhesion to endothelial cells, and the release of lysosomal enzymes, preventing chemotaxis at the site of inflammation. Finally, it increases IL-10 gene expression, thus exhibiting further anti-inflammatory effects [57].

Another important function of dexamethasone is its implication in the process of the resolution of inflammation by inducing the D-series pro-resolving lipid mediator pathway via the formation of protectins D1 and DX7 and of 17S-dihydroxydocosahexaenoic acid (17S-HDHA) [58]. These agents have documented roles in driving the resolution of inflammation by regulating the termination of proinflammatory responses, the removal of inflammatory cells, and the tissue repair mechanisms [59]. Finally, certain lipid mediators have been shown to diminish pathologic thrombosis and NETosis, decrease the levels of procoagulant factors and fibrinogen, enhance the removal of clots, and increase the anticoagulant factors [60]. These functions are of high importance, since the thrombotic pathways are central in the pathology of COVID-19 infection.

Overall, in critical COVID-19 patients, glucocorticoid therapy exerts several functions, such as effective homeostatic corrections and consequent improvement in the critical illness-related corticosteroid insufficiency (CIRCI), control of inflammation, and stabilization of cardiovascular and metabolic functions. Furthermore, it induces increased mitochondrial biogenesis, corrects the content and functions of the mitochondria, and decreases the devastating tissue oxidation that leads to multiple tissue and organ dysfunctions [61].

## 4. Dexamethasone in COVID-19: Clinical Considerations

Starting with the main results of the RECOVERY trial, the most robust until now, that gave dexamethasone a central role in the treatment of COVID-19, dexamethasone at a dose of 6 mg per day for ten days was effective at reducing the mortality rate in both ventilated and oxygenated patients by one-third and one-fifth, respectively, compared to the usual care alone [16].

However, several questions relative to corticosteroid therapy arose:1.Why use dexamethasone instead of other corticosteroids?2.When to start corticosteroids?3.At what dose and for how long?4.Do they prolong viral shedding?5.How safe are corticosteroids in the treatment of severely affected COVID-19 patients?6.What is the role of inhaled corticosteroids in preventing or treating COVID-19 patients?7.Why use dexamethasone instead of other corticosteroids?

Pharmacologically, dexamethasone has a more advantageous pharmacokinetic/pharmacodynamic (PK/PD) profile. Dexamethasone is a long-acting synthetic corticosteroid with systemic effects and is almost 25 times more potent than other synthetic corticosteroids, as depicted by the equivalent steroid doses (40-mg hydrocortisone:10-mg prednisolone:8-mg methylprednisolone:1.5-mg dexamethasone) [62].

Dexamethasone’s higher potency is explained by its higher affinity for the glucocorticoid receptor compared to other steroids, leading to a more profound genomic and metagenomic activity [62,63,64] and consequent greater inhibition of the proinflammatory pathways. Additionally, it possesses minimal mineral corticoid activity, limiting the undesired retention of the fluid and sodium. On the PK side, dexamethasone has a longer half-life of 36–72 h, allowing for once-daily dosing [65].

### 4.1. When to Start Corticosteroids?

The time of corticosteroid treatment initiation is of great importance in order to have favorable outcomes. Siddiqi and Mehra [66] suggested three escalating phases of COVID-19 disease progression, with associated signs, symptoms, and potential phase-specific treatments: the early infection phase, the pulmonary phase, and the hyperinflammation phase, in which anti-inflammatory treatments (such as corticosteroids) are recommended as beneficial. Indeed, in the RECOVERY trial [16], a clear benefit was shown in the patients who were treated with dexamethasone for more than 7 days after symptom onset, when inflammatory lung damage was likely to be more common, and not among those with more recent symptom onsets. It is evident that randomized controlled trials are needed in order to clarify the right timing for the initiation of a corticosteroid treatment course.

### 4.2. At What Dose and for How Long?

Another consideration is whether “one size fits all” and how effective dexamethasone is in the dose of 6 mg/day for ten days in all COVID-19 patients with respiratory failure, irrespectively of them developing ARDS or not. This daily dose is considered low, with few adverse events [39], as opposed to the 20–40 mg needed daily for hematologic malignancies, autoimmune diseases, or shock.

In an early single-center observational study in Wuhan [61], there was a lower risk of death in the patients who received low-dose corticosteroids (<1 mg/kg/day) or no corticosteroids at all compared to high-dose regimens, but the difference was not statistically significant.

In the CoDEX randomized trial in patients with COVID-19 with moderate-to-severe ARDS, the dose of dexamethasone administered (20 mg of dexamethasone intravenously daily for 5 days and 10 mg of dexamethasone daily for 5 days or until ICU discharge) [27] was higher than that used in the RECOVERY trial. The selection of this higher dose was based on the design of a previous study [67] in patients with non-COVID-19 ARDS who were treated with dexamethasone (or equivalent) in the dose of 30 mg/day. Eventually, a statistically significant increase in the number of ventilator-free days (days alive and free of mechanical ventilation) was shown over 28 days, with a good safety profile [27].

Furthermore, Papamanoli et al. [43] showed that, in nonintubated patients with severe COVID-19 pneumonia, higher doses of methylprednisolone (160 mg for a median duration of approximately 10 days) were associated with a lower risk for mechanical ventilation or death (37%) and without consequent adverse effects. Interestingly, among the intubated patients, those that were treated with corticosteroids had more ventilator-free days and shorter length of ICU stays compared to the patients not receiving corticosteroids.

Fernandez-Cruz et al. [32] showed lower in-hospital mortalities in patients with SARS-CoV-2 pneumonia treated with corticosteroids, no matter if they received an initial regimen of 1 mg/kg/day of methylprednisolone or with corticosteroid pulses.

A more recent review and meta-analysis of 35 studies by Cano et al. [26] depicted the high heterogeneity of the practices in dosing steroids; in 74.2% of the studies, steroids were given in low doses, in 11.4%, in high or pulse doses, and in 5.6%, a mixed regimen was used. The meta-analysis failed to prove a beneficial or harmful effect of either dosing scheme.

To complicate the dosing dilemmas further, the published data showed that serum albumin plasma levels, competing drugs, and albumin glycation are important clinical variables influencing the effectiveness of dexamethasone in treating COVID-19 patients [68]. Each one of these factors decreases the binding capacity of albumin, complicating the transport capacity of dexamethasone and resulting in a shorter half-life of the drug and potential toxicity at its peak concentration. It might be appropriate to modify the dose of dexamethasone according to the albumin levels, patient’s weight, and/or plasma levels in order to achieve the therapeutic levels and to avoid toxicity [69].

From a pathophysiology point of view, as Chrousos and Meduri concluded, corticosteroid administration in severe COVID-19 should start early, before the irreversible exhaustion of homeostatic reserves, with large doses to saturate the ubiquitous glucocorticoid receptors, so that the maximum effect is attained [70,71]. Following precision medicine, the dosing of corticosteroids should be tailored to each patients’ unique manifestations, as Gogali et al. explained [72]. From the evidence-based medicine side, though, it is evident that the optimal corticosteroid dosing and duration of treatment still need clarification, especially in critically ill COVID-19 patients with ARDS. Towards this direction, the results of currently undergoing trials like “HIGHLOWDEXA” [20], assessing the efficacy of high doses of dexamethasone (20 mg/day for 5 days and then 10 mg/day for 5 days) vs. low doses of dexamethasone (6 mg/day for 10 days) in patients with respiratory failure might shed more light on the therapeutic algorithms of severe COVID-19.

### 4.3. Do Corticosteroids Prolong Viral Shedding?

An analysis of the retrospective observational studies from the first pandemic wave showed results associating corticosteroids with immune suppression, an increased viral load, delayed clearance from the body, and increased mortality [21,22]. The early administration of corticosteroids in non-severely ill patients may be deleterious due to an increase of viral shedding or a delay in viral clearance [46]. Prolonged viral shedding has been supported by several retrospective studies [14,36]. Notably, nonsevere COVID-19 pneumonia patients who received early corticosteroid treatment had a more prolonged viral shedding compared to patients of similar severity who did not receive corticosteroids, as was shown in two cohorts [47,48] and one randomized study [42]. A recent review and meta-analysis of 15 studies examining, among others, the SARS-CoV-2 clearance concluded a delay in viral clearance in patients receiving corticosteroids compared with patients who did not receive corticosteroids but failed to prove a statistical significance [20]. Importantly, there are implications that viral shedding prolongation follows a dose-dependent scheme of corticosteroids [49]. However, this theory was not confirmed in the study of Fang et al. [51], who used a low dose of corticosteroids in patients with COVID-19; there was no significant difference in the duration of viral shedding between patients receiving and patients not receiving corticosteroids, irrespective of the severity of the disease. Accordingly, no significant difference in the viral clearance was shown in other studies [22,30,37,38,45]. Two retrospective studies by Xu, K. and Shi, D. [73,74] concluded that corticosteroids were not independent predictors of prolonged viral shedding as opposed to the male sex, lymphocytopenia, APACHE score, and IVIG treatment.

To sum up, there is no clear conclusion as to whether corticosteroids delay the SARS-CoV-2 clearance, and more importantly, there is no certainty about the clinical significance of this.

### 4.4. How Safe Are Corticosteroids in the Treatment of Severely Affected COVID-19 Patients?

The long-term use of corticosteroids has well-established adverse effects. The short-term use of steroids, although frequent in the pre-pandemic era, was only recently examined for its potential adverse effects. Two large population-based studies in adults, an American [50] and a Chinese [75] one, underline the increased risk of complications in the outpatient setting. Waljee et al. [50] concluded that, within 30 days of steroid initiation, there was a two to five-fold increase in the incidence rates of fractures, venous thromboembolism, and admission for sepsis, in that order of frequency, in steroid users 18–64 years of age compared to non-users. Yao et al. [75] found a significant increase in gastrointestinal bleeding, sepsis, and heart failure in the respective Chinese population.

Currently the recommendations on corticosteroid use in COVID-19 concern severely ill inpatients requiring O_2_ or mechanical ventilation and definitely not outpatients. Is this sicker population threatened by the same harms of short-term corticosteroids use as the general population? The existing literature on COVID-19, although abundant, is not centered on adverse events.

Certain studies concluded that there were no significant differences in serious adverse events between corticosteroid and the standard-of-care groups [24,27,37,42,45,46], while others pointed to a greater probability of myocardial or liver injury, shock [21], and agitation [23] in corticosteroid-treated COVID-19 patients.

Supporting the safety of corticosteroids, the RECOVERY trial [16] reported a similar incidence of new cardiac arrhythmia in the dexamethasone group and the usual care group. In a total of 2104 patients, there were four reports of serious adverse reactions that were deemed by the investigators to be related to dexamethasone: two of hyperglycemia, one of gastrointestinal hemorrhage, and one of psychosis (all recognized adverse effects of corticosteroids).

In the CAPE COVID trial [28], which was stopped early after the publication of the results from the RECOVERY trial, the administration of a low dose hydrocortisone in 148 patients was followed by three serious events: one was cerebral vasculitis, attributed to the SARS-CoV-2 itself, the second was an episode of pulmonary embolism with cardiac arrest, which was considered single and unrelated to steroids, and the third event was an intrabdominal hemorrhage related to anticoagulation for pulmonary embolism.

In REMAP-CAP [29], the only adverse events that were considered relevant to the corticosteroid treatment were one event of neuromyopathy and one event of fungemia.

The recently published results of the OUTCOMEREA network [76] showed that COVID-19 critically ill patients have a higher risk for bloodstream infections (BSI) than their non-COVID-19 counterparts after 7 days of ICU stay, but this negative event was associated with the use of anti-IL-1 and anti-IL-6 and not to corticosteroids. Similarly, no difference in BSI incidence was detected in the MetCOVID randomized trial [30] or the retrospective study by Papamanoli et al. [43].

On the contrary, Giacobbe et al. [77] in their retrospective single-center study, found a higher BSI risk for corticosteroid-treated patients (csHR 3.95, 95% CI 1.2–13.03) than those receiving only tocilizumab (csHR 1.07, 95% CI 0.38–3.04), the risk being even higher with their combination (csHR 10.69, 95% CI 2.71–42.17).

The clinical research points towards an association between corticosteroid use and secondary infections. More secondary infections with sepsis have been recorded [26,36], but the difference was not always significant [13]. The indirect evidence on increased secondary infections may come from the increased antibiotic use in corticosteroid-treated patients, as two retrospective studies [47,48] and one systematic review and meta-analyses [26] have shown.

The literature is more extensive on fungal infections in COVID-19 patients. Corticosteroid treatment is a well-known risk factor for invasive mycoses [78]. An excellent review of retrospective studies from China and Europe reported a 20–35% incidence of COVID-19-associated pulmonary aspergillosis (CAPA) [79] A case report of mucormycosis following the treatment with dexamethasone was reported in a 44-year-old diabetic woman [80] raising concerns about the duration and dosing of the corticosteroid treatment in diabetic COVID-19 patients. Hyperglycemia, which led to increased doses of insulin, was mentioned in most studies [13,23,30,42] as an adverse event of the corticosteroid treatment as expected, but it was not severe [27].

Reviewing the literature concerning the adverse events of corticosteroid treatment in COVID-19 patients, we underline a significant deficit, as safety is not among the primary outcomes of most studies. This is not a surprise, because corticosteroids are used extensively, and physicians are very familiar with their use. Additionally, some of the main adverse effects of corticosteroids (i.e., VTE, secondary infections and GI hemorrhage) are the main COVID-19 complications or harms from other necessary treatments (i.e., anticoagulation), and perhaps, it is difficult to discriminate the proportional contribution of the virus or the steroids.

Targeting a safe treatment alternative to systemic corticosteroids, inhaled corticosteroids (ICS) have been proposed. There is a cumulative experience of ICS in chronic airways diseases like asthma and COPD, linking their use to changes in the microbiome and, finally, to the increased frequency of upper and lower respiratory tract infections [81,82]. However, an association of ICS with the susceptibility to viral infections is lacking. Instead, a paradoxical protection against viral infections has risen [83,84]. Ciclesonide seems to block SARS-CoV-2 replication in vitro, and in addition, it impedes its cytopathogenicity [85,86]. Early-stage COVID-19 treatment with inhaled corticosteroids might inhibit the progression of SARS-CoV-2 infection to COVID-19. This was shown in a small study of COVID-19 patients on oxygen (not mechanically ventilated) who improved after the treatment with inhaled ciclesonide [87] Similarly, there was in vitro evidence of budesonide’s direct antiviral and anti-inflammatory properties [88]. In the recent open-label, in parallel groups, phase 2 randomized control trial in mild outpatient COVID-19 patients that received inhaled budesonide vs. the usual care, a 91% reduction in the relative risk for COVID-19-related urgent care or hospitalization was shown in the budesonide-treated patients [89].

Patients with chronic lung disease, often on long-term ICS, form a special subcategory of COVID-19 patients. A systematic review of the patients with coronavirus acute respiratory infections (MERS, SARS and COVID-19) by Halpin et al. [90] did not reach a clear-cut conclusion as to whether premorbid use or the continuous administration of ICS is a benefit. Currently, there is a suggestion against the withdrawal of chronically used ICS; asthma and COPD patients receiving ICS are advised to carry on with their treatments [91,92]. Table 5 summarizes the commonest reported adverse events.

## 5. Conclusions

In conclusion, dexamethasone has gained a major role in the therapeutic algorithm of patients with COVID-19 pneumonia requiring supplemental oxygen or on mechanical ventilation. Its wide anti-inflammatory action seems to form the basis for its beneficial action, taming the overwhelming “cytokine storm”. Amid a plethora of scientific research on the therapeutic options for COVID-19, there are still unanswered questions about the right timing, right dosing, and right duration of corticosteroids. Lastly, since medicines treat patients, we urgently need more evidence on the safety of corticosteroids on specific patient populations who may be at risk for more complications.

## Figures and Tables

**Table 1 jpm-11-00556-t001:** Trials using methylprednisolone (MP); by design (R = randomized in blue and Rs = retrospective and obs = observational in yellow), timing, duration, outcomes (focusing on mortality data), and viral shedding.

Trial Name/Authors (Reference)	Steroid Used, Dosing	Trial Design, Population	Initiation Timing (Days)	Duration of Administration (Days)	Main Outcome(s)	Secondary Outcome(s)	Viral Clearance
Jeronimo CMP et al. MetCOVID [30]	MP 0.5 mg/kg × 2 vs. placebo	MC/R/DB phase II-b placebo-controlled		5	No difference in D28 mortality MP pts > 60 yo had lower D28 mortality	No difference in: D7, D14 mortality/D7 intubation and PFR < 100/LOHS/Radiological presence of fibrosis/BOOP after D7	No difference on D7viral clearance
Corral-Gudino L et al. GLUCOCOVID trial [13]	MP 40 mg bid × 3 d, then 20 mg bid × 3 d	MC/O-L/R pts on O_2_ SOC vs. SOC + MP	NA	6	Composite endpoint: death, ICU admission, need for NIV MP: no significant effect on endpoint (ITT analysis) MP: beneficial effect (PP analysis)	NA	NA
Tang X et al. [42]	MP 1 mg/kg/d vs. no MP	Ps/MC/R Single-blind ward pts	8 (6–16) since Sx onset	7	MP vs. no MP: no difference in incidence of clinical deterioration (4.8 vs. 4.8%; OR 1.000 [95% CI, 0.134–7.442]; *p* = 1.000)	MP vs. no MP: no difference in: D14 clinical cure rate/time to clinical cure/ICU admission/hospitalization duration/in-hospital mortality (all *p* > 0.05)	MP significantly prolonged SARS-CoV-2 shedding (median, 11 d vs. 8d; HR 1.782 (1.057–3.003); *p* = 0.030)
Gong Y et al. [21]	MP 1 to 2 mg/kg/d halved every 3 d vs. no MP	Rs		5–10	No difference in radiologic progression within 20 d		Longer time to negative PCR in MP group (*p* = 0.03)
Wang et al. [35]	MP 1 to 2 mg/kg/d	Obs severe COVID-19 pts		5–7	MP: faster SpO_2_ improvement, less likely to receive MV (*p* = 0.05), faster ↓ CRP, IL-6 No significant difference in: mortality, ↓ LOHS and ICU LOS		
Papamanoli A et al. [43]	MP median 160 mg (120–180)/d	SC/Rs cohort HFNC > 50%, MP vs. no CS	10 since Sx onset, 2 since admission, 1 since HFNC initiation	Median: 10 incl tapering	MP: 37% lower risk of death D28 (*p* = 0.003) and less frequent MV (*p* = 0.001)	No difference in mortality between groups	
Fernandez-Cruz et al. [32]	MP 1 mg/kg/d or pulse	SC/Rs MP vs. no MP	10 since Sx onset	NA	MP Mortality 13.9% vs. 23.9% no MP (HR = 0.51 95% CI, 0.27–0.96, *p* = 0.044). Dosing scheme not associated with mortality.	Mortality in moderate-severe ARDS: MP 26.2% vs. 60% no CS, OR = 0.23 (95% CI, 0.08–1.71)	
Zha L et al. [44]	MP 40–80 mg/day	Obs/MC CS vs. no CS	within 24 h after admission	5	LOHS and Duration of Sx: not associated with CS		CS: no influence on viral clearance
Li Y et al. [45], Shangai cohort	MP 0.75–1 mg/kg/d × 3 d, then 20 mg MP × 3 d vs. (no MP and rescue CS)	MC/Obs/	Early (according to LDH and radiographic progression)	≤7	↓ MV need in early MP (*p* = 0.037)		No difference in viral clearance time
Ma Q et al. [38]	MP 40–80 mg/d vs. no CS	MC/Rs cohort severe and critically ill pts			No difference: Mortality, LOHS CS group: ↓ Sx duration		No difference in viral clearance time
Salton F et al. [22]	MP 80 mg loading dose, then 80 mg/day cont infusion	MC/Obs severe COVID-19 MP vs. no MP		Min 8	Composite endpoint:ICU referral/ intubation need/ D28 death: significantly ↓ compared with the control group: aHR, 0.41	MV free d ΔCRP	Not affected
Yuan M et al. [46]	MP max dose 52.5 mg	Rs nonsevere pts MP vs. no MP	Median 8.3 since Sx onset	Median duration 10.8	Nonsevere pts on CS:progressed to severe disease, had ↑ LOHS and ↓c duration of fever (not statistically significant difference vs. non-CS pts)		Nonsevere pts on CS: ↑ duration of viral shedding, (not statistically significant difference)
Wu C et al. [31]	MP vs. no MP	Rs/Obs	NA	NA	MP in ARDS pts: ↓ risk of death (HR, 0.38; 95% CI, 0.20–0.72, *p* = 0.003)		NA

**Table 2 jpm-11-00556-t002:** Trials using dexamethasone (dexa); design (R = randomized in blue and Rs = retrospective and obs = observational in yellow), timing, duration, outcomes (focusing on mortality data), and viral shedding.

Trial Name/Authors (Reference)	Steroid Used, Dosing	Trial Design, Population	Initiation Timing	Duration of Administration (Days)	Main Outcome(s)	Secondary Outcome(s)	Viral Clearance
Tomazini BM et al. CoDEX trial [27]	dexa 20 mg iv × 5 d, then 10 mg dexa × 5 d or until ICU discharge	MC/R/O-L moderate to severe ARDS: SOC vs. SOC + dexa	Not mentioned	≤10	Dexa group: mean 6.6 ventilator-free d (95% CI, 5.0–8.2) vs. SOC group: 4.0 ventilator-free d (95% CI, 2.9–5.4), (difference, 2.26; 95% CI, 0.2–4.38; *p* = 0.04).	D28 no significant difference in: All-cause mortality Clinical status ICU free d MV duration SOFA	NA
RECOVERY trial [16]	dexa 6 mg/day pos or iv	Controlled/O-L CS vs. SOC	NA	10	D28 mortality: 22.9% dexa vs. 25.7% SOC, greater ↓ in pts on MV > O_2_ and if initiation > 7 d	Time to discharge Need for MV Need for CRRT	NA

**Table 3 jpm-11-00556-t003:** Trials using hydrocortisone (HC); design (R = randomized in blue), timing, duration, outcomes (focusing on mortality data), and viral shedding.

Trial Name/Authors (Reference)	Steroid Used, Dosing	Trial Design, Population	Initiation Timing	Duration of Administration (Days)	Main Outcome(s)	Secondary Outcome(s)	Viral Clearance
Dequin P-F et al. [28]	HC Cont iv: 200 mg × 7 d, then 100 mg × 4 d, then 50 mg × 3 d	MC/R/DB critically ill pts		Total max 14	D21 Tx failure (death, persistent MV, HFNC): No significant difference between CS and placebo (under-powered, stopped early)	No significant difference in: need for intubation, proning, ECMO, NO	
Angus DC et al. [29]	HC Fixed: 50–100 mg × 4 Shock HC: 50 mg × 4	O-L/R, severe COVID-19 pts, fixed dose HC vs. shock dose HC vs. no HC		7	D21 organ-support free ds: Probability of superiority compared to no CS: 93% in fixed dose HC 80% in shock dose HC	Mortality rate: 30% fixed dose 26% shock dose 33% no CS	

**Table 4 jpm-11-00556-t004:** Trials using different corticosteroids (CS); design (Rs = retrospective and obs = observational and As = ambispective in yellow and MA = meta-analysis and SRMA = Systematic Review and Meta-analysis in orange), timing, duration, outcomes (focusing on mortality data), and viral shedding.

Trial Name/Authors (Reference)	Steroid Used, Dosing	Trial Design, Population	Initiation Timing	Duration of Administration (Days)	Main Outcomes	Secondary Outcome(s)	Viral Clearance
Wu C et al. [14]	Max dose: 80 mg MP equiv (IQR = 40–80) mg	SC/Rs/Obs	NA	7 (4–12)	D-60 in-hospital-mortality: Significant ↓ in CS (*p* = 0.0160)	NA	CS not associated with delayed viral clearance
Liu J et al. [18]	Not mentioned	Rs/Obs CS vs. no CS			CS D28 mortality 44.3% vs. no CS 31% (*p* < 0.001) Mortality in CS group linked to HD and early initiation (<3 d from hospitalization)		Delayed (SHR 1.59 CI 95%, 1.17–2.15) *p* = 0.003
Li Y et al. [36]	Median 200 mg/d HC equiv (100–320.9)	Rs/MC critically ill pts CS vs. no CS		9 (5–14)	No significant difference in mortality		CS prolonged viral shedding
Lu X et al. [19]	Different CS, 100–800 mg/day HC equiv	SC/Rs, critically ill pts, CS vs. no CS		Median 8 (4–12)	CS independent from overall mortality. HD CS:↑ mortality risk (*p* = 0.003)	NA	NA
Li Q et al. [47]	5 pts on pos prednisone (MP equiv dose 20 mg/d) and 50 pts on iv MP 20–40 mg/d	Rs nonsevere pts early LD CS vs. no CS	1–5 d after hospital admission	3 (Oral prednisone) 3–5 (iv MP)	CS in nonsevere COVID-19 pneumonia: ↑risk of progression to severe disease, ↑ use of antibiotics, ↑ duration of fever, ↑LOHS		CS in nonsevere COVID-19 pneumonia: prolongs virus clearance time
Ma Y et al. [48]	MP equiv 56.6 mg median daily dose	Rs/MC, severe and nonsevere COVID-19 pts		5 median duration	CS: ↑ LOS in nonsevere pts (*p* < 0.05) BUT no significant ↑ LOS of severe pts		CS:↑ viral shedding duration only in nonsevere pts (*p* < 0.05)
Li S et al. [49]	Not mentioned		Not mentioned	Not mentioned	Assessment of risk factors for long-term (>30 d) positive SARS-CoV-2 and viral shedding	NA	HD (80 mg/day; aHR, 0.67 (95%CI, 0.46–0.96) *p* = 0.031) but not LD CS (40 mg/day; aHR,0.72 (95%CI, 0.48–1.08), *p* = 0.11) potentially delayed viral shedding
Shi D et al. [50]		Rs/SC			Factors associated with prolonged viral shedding		CS:not independent factor of prolonged viral shedding
Hu Y et al. [37]	0.75–1 mg/kg/d MP equiv	Rs CS vs. no CS	Median time since Sx onset: 7	6 (IQR, 4–8)	CS vs. no CS: no significant difference in: imaging progression, 90-day mortality. Significant difference in lymphocytes and CRP		CS vs. no CS: no significant difference in time to negative PCR
Keller MJ et al. [33]	CS vs. no CS	Obs incl children	Within 48 h of admission		CS: no effect on mortality, BUT if CRP > 20; significant ↓ risk of mortality or MV [OR 0.23, 95% CI, 0.08–0.7]. If CRP < 10; significant ↑ risk of mortality or MV [OR 2.64, 95% CI, 1.39–5.03]		
Wu J et al. [40]	MP equiv dose 40 mg/d	Rs severe and critically ill pts	24 h after Dx	Median 6 (3–10)	CS independently associated with ↑ in-hospital mortality in severe COVID-19 (HR = 1.43, 95% CI: 0.82–2.49, *p* = 0.201 in IPTW; HR = 1.55, 95% CI: 0.83–2.87, *p* = 0.166 in PSM) and critically-ill pts (HR = 3.34, 95% CI: 1.84–6.05, *p* < 0.001 in IPTW; HR = 2.90, 95% CI: 1.17–7.16, *p* = 0.021 in PSM)		
Albani F et al. [39]	Median dexa equiv dose 20 mg/day	Rs CS vs. no CS			CS not associated with in-hospital mortality	CS: ↓ ICU admission	
Fang X et al. [51]	Median dose (HC equiv) pos 237.5 mg/iv 250 mg in general/severe pts	Rs, pts of different severity, CS vs. no CS	Not mentioned	Median duration: 7/4 in general/severe pts			CS:no significant difference in viral shedding irrespective of severity
Li X et al. [41]	Median cumulative dose equiv to 200 mg prednisone	As, CS vs. no CS	NA	Median duration: 4	severity on admission, complications, Tx, and outcomes: HD CS during hospitalization (aHR, 3.5; 95%CI, 1.8–6.9] was significant risk factor associated with death in severe COVID-19	NA	NA
Sarkar S et al. [20]		SRMA (2 RCTs and 10 cohorts)			CS: ↑ mortality (OR = 1.94, 95%CI: 1.11–3.4, *I*^2^ = 96%), irrespective of severity or dosage and ↑ LOHS (MD = 1.18 d, 95%CI: −1.28 to 3.64, *I*^2^ = 93%)		CS may prolong viral shedding (MD = 1.42 d, 95%CI: −0.52 to 3.37, *I*^2^ = 0%)
Sterne JAC et al. REACT [23]	Different CS in different schemes	Ps/MA (7 RCTs) CS vs. SOC			CS: ↓ all-cause D28-mortality: dexa vs. SOC OR 0.64 [95% CI, 0.5–0.82. *p* < 0.01], HC vs. SOC OR 0.69 [95% CI, 0.53–0.82, *p* = 0.13], MP vs. SOC OR 0.91 [0.29–2.87, *p* = 0.87]	No difference in serious AEs	
Pasin L et al. [24]		SRMA (5 RCT)CS vs. any other Tx			CS: ↓ Mortality rate (26% in CS vs. 28% in no CS, RR = 0.89, CI 95%, 0.82–0.96, *p* = 0.003) CS: ↑ mortality in pts on no O_2_	Significant ↓ of MV risk in CS group	
Van Paassen [25]	Diverse CS strategies	SRMA (RCTs and Obs)			CS are beneficial in short-term mortality	CS: ↓ MV need	CS: not clear effect on viral clearance
Cano EJ et al. [26]	Different CS in different doses	SRMA			Mortality benefit in CS group in severe COVID-19: OR, 0.65; 95% CI, 0.51–0.83, *p* = 0.0006. No benefit or harmful effect among HD and LD CS		LD CS no significant effect on viral shedding

AE = Adverse Events, aHR = adjusted hazard ratio, ARF = acute respiratory failure, As = ambispective, CI = confidence interval, cont = continuous, CS = corticosteroids, DB = double-blind, dexa = dexamethasone, Dx = diagnosis, equiv = equivalent, HC = hydrocortisone, HD = high-dose, HR = Hazard Ratio, incl = including, IPTW = inverse probability of treatment weighting, IQR = interquartile range, ITT = Intention-to-treat, LD = low-dose, LO(H)S = Length of (hospital) stay, MA = meta-analysis, MC = Multicenter, MP = methylprednisolone, MV = mechanical ventilation, NA = not applicable, NIV = noninvasive ventilation, Obs = observational, O-L = Open-label, OR = odds ratio, pos = per os, PP = per protocol, Ps = prospective, PSM = propensity score matching, pts = patients, R = Randomized, RCT = Randomized-controlled trial, RR = relative risk, Rs = Retrospective, SC = single center, SHR = subhazard ratio, SI = systemic inflammation, SOC = standard of care, Sx = symptom(s), SRMA = Systematic Review and Meta-analysis, Tx = treatment, vs. = versus, yo = years old, and Δ = change.

**Table 5 jpm-11-00556-t005:** Trials and corticosteroid (CS)-related adverse effects (randomized trials in blue, retrospective/observational trials in yellow, and meta-analysis in orange).

Trial Name	Steroid Used, Dosing	Initiation (Days)	Duration (Days)	Hyperglycemia	GI	CNS	CVS	Secondary Infections	Other
Jeronimo CMP et al. MetCOVID [30]	MP 0.5 mg/kg bid vs. placebo		5	MP group: ↑insulin				No difference in BC positivity	
Corral-Cudino et al., GLUCOCOVID [13]	MP 40 mg bid × 3 d, then 20 mg bid x 3 d	Min 7 after Sx onset	6	MP: ↑ Glu (*p* = 0.015)				MP: ↑ Secondary infections (*p* = 0.637)	
Tang X et al. [42]	1 mg/kg/d MP vs. no MP	Median time 8 (6–16) after Sx onset	7	no MP: ↑ Glu (*p* = 0.313)	Either group: No stress ulcers/GI bleed	Either group: No delirium		MP: ↑ VAP (*p* = 0.557)	
Papamanoli A et al. [43]	Median daily 160 mg MP (120–180)	10 from Sx onset, 2 from admission, 1 from HFNC initiation	Median: 10 incl tapering		GI bleed: no difference			Bacteremia, HAP/VAP: No difference	
Salton F et al. [22]	80 mg MP loading dose, then 80 mg/day cont infusion		Min 8	MP: ↑ Glu		MP: Mild agitation more common			
Yuan M et al. [46]	MP max dose 52.5 mg, nonsevere pts	Median 8.3 from Sx onset	Median duration 10.8					CS vs. no CS: No significant difference in secondary infections	
RECOVERY trial [16]	dexa 6 mg/day pos or iv			2/2104 pts	GI bleed: 1/2104 pts	Psychosis: 1/2104			
Tomazini BM et al. CoDEX trial [27]	20 mg dexa iv × 5 d, then 10 mg dexa × 5 d or until ICU discharge		≤10	Insulin need: Dexa: 31.1% vs. SOC 28.3%				Dexa 21.9% vs. SOC: 29.1%	Other serious: dexa 3.3% vs. 6.1% SOC
Dequin P-F et al. [28]	Cont iv: 200 mg HC × 7d, then 100 mg × 4d, then 50 mg × 3 d		Total max 14					D28: nosocomial infection: 37.3% HC vs. 41.1%placebo (*p* = 0.42). D28 VAP: 29% HC vs. 27.4% placebo. D28 bacteremia: 6.6% HC vs. 11% placebo	HC:3 serious AE considered unrelated: Cerebral vasculitis, Cardiac arrest 2nd to PE, IA bleed 2nd to anticoagulation
Angus DC et al. [29] REMAP-CAP COVID-19	Fixed HC: 50–100 mg qid vs. Shock HC: 50 mg qid vs. No CS (101 pts)		7					1 episode of fungemia in fixed dose HC	1 episode of neuromyopathy in fixed dose HC
Li Y et al. [36]	Median 200 mg/d HC equiv		9 (5–14)					CS: ↑ Secondary infections	
Liu J et al. [18]	Not mentioned				Liver injury: CS 18.3% vs. 9.9% no CS (*p* = 0.001)		myocardial injury: CS 15.6% vs. 10.4% no CS (*p* = 0.041)		Shock: CS 22% vs. no CS 12.6% (*p* < 0.001)
Li Q et al. [47]	Pos Prednisone (MP equiv dose 20 mg/d) or iv MP 20–40 mg/d	1–5 after hospital admission	Pos prednisone × 3, iv MP × 3–5					CS in nonsevere COVID-19 pneumonia: ↑ use of antibiotics	
Ma Y et al. [48]	CS 56.6 mg (MP equiv) median daily dose		5 median duration					CS: ↑ antibiotic use regardless of disease severity (*p* < 0.001)	
Hu Y et al. [37]	0.75–1 mg/kg/d MP equiv	Median since Sx onset: 7	6 (IQR, 4–8)	No difference					No difference in hypokalemia
Li Y et al. [45]	MP 0.75–1 mg/kg/d × 3 d, then 20 mg × 3 d vs. (no MP and rescue CS)	Early (according to LDH and radiographic progression)	≤7			No difference in psychosis		No difference in secondary infections	No difference in osteoporosis, avascular necrosis
Buetti N et al. [77]	Matched case- control study COVID-19 vs. non-COVID-19 ICU pts							COVID-19 pts:↑ BSI probability, esp after 7 d of ICU admission compared to non-COVID-19 ICU pts (*p* < 0.00001). COVID-19 pts: significantly ↑ ICU-BSI risk if received anti-IL-1 or anti-IL-6 (sHR 3.20, 95% CI 1.31–7.81, *p* = 0.011) but not in pts who received CS.	
Sterne JAC et al. REACT [23]	Different CS in different schemes								No difference in serious AEs
Van Paassen [25]	Diverse CS strategies		5–10					CS: ↑ use of antibiotics and ↑ secondary infections or sepsis	

BC = blood culture, BSI = bloodstream infections, esp = especially, GI = gastrointestinal, ↑ Glu = hyperglycemia, IA = intra-abdominal, PE = pulmonary embolism, and VAP = ventilator- associated pneumonia. CNS=central nervous system; CVS: cardiovascular system.

## Data Availability

Not applicable.

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
