# Peer review of "Dexamethasone in the Treatment of COVID-19: Primus Inter Pares?"

_jpm, 2021, doi:10.3390/jpm11060556_

Round 1

Reviewer 1 Report

The review is well documented and well organized. The authors adress major issues regarding the use of dexamethasone in the treatment of covid-19. I feel though that english language must be reviewed and some paragraphes need rephrasing. 

Author Response

Reviewer 1

The review is well documented and well organized. The authors address major issues regarding the use of dexamethasone in the treatment of covid-19. I feel though that english language must be reviewed and some paragraphes need rephrasing. 

Response: 

We would like to thank the reviewer for the positive comments. A thorough editing of the manuscript has been made, as suggested. Long paragraphs have been rephrased and re-written. A careful editing to the English language has been done as well, with the appropriate corrections where needed.

Reviewer 2 Report

Dear Authors,

The manuscript by Romanou V et al, entitled “Dexamethasone in the treatment of COVID-19: primus inter pares?” is an extensive review on: Coronavirus disease 2019 (COVID-19), caused by severe acute respiratory syndrome coronavirus 2 (SARS-CoV2), rapidly spread around the world since its offspring. Although most patients present with mild symptoms about 15% of COVID-19 cases become severe, rapidly developing acute respiratory distress syndrome (ARDS) and subsequent increased risk of death. Corticosteroids have been used for the treatment of COVID-19 patients as they are reported to elicit benefit by reducing lung inflammation, and inflammation-induced lung injury. Dexamethasone has gained a major role in the therapeutic algorithm of patients with COVID-19 pneumonia requiring supplemental oxygen or on mechanical ventilation. Aim of this review was to summarize the data on dexamethasone treatment in COVID-19 and outline the clinical considerations of corticosteroid therapy in these patients.

The authors have performed extensive bibliographic research of current literature on a timely subject, however, probably due to the language barrier, many sentences and ideas are not clear or easy to understand. Additionally, some sentences are too long. The manuscript needs major revisions.

  1. Line 45: Add reference at the end of “61.5% around the world”.
  2. Lines 71-74: Rephrase. “Still, ……….
  3. Line 153: correct highly to high
  4. Lines 155-160: Rephrase. “In the overall, ………. dysfunction (51).
  5. Line 168: Rephrase. What do you mean?
  6. Line 169: Rephrase using a verb.
  7. Line 170: Rephrase.
  8. Use the above corrected subtitles (where appropriate) in the rest of the document.
  9. Line 192-196: Rephrase. “Siddiqi ………. onset.
  10. Lines 214-216: Rephrase.
  11. Lines 222-224: Rephrase.
  12. Lines 327-332: Rephrase.
  13. Lines 347-352: Rephrase.
  14. When were references 10 (lines 407-408) and 64(lines 548-549) accessed?
  15. In table 1 replace all the abbreviations used in the table (many of which are miss-spelt) to gain space and homogeneity.
  16. In table 1 organize the enlisted articles according to: (a) which steroid was used. For example, all Dexa reports together etc. (b) Trial design, thereafter. For example, all Retrospective trials together etc.

Author Response

Reviewer 2

We would like to thank the reviewer for the careful reading of our manuscript and the valuable suggestions for improving the meaning in several parts of the paper.

  1. Line 45: Add reference at the end of “61.5% around the world”.

Response

References have been added as suggested.

  1. Lines 71-74: Rephrase. “Still, ……….

Response

The paragraph has been rephrased as suggested by the reviewer.

  1. Line 153: correct highly to high

Response

The correction is done

  1. Lines 155-160: Rephrase. “In the overall, ………. dysfunction (51).
  2. Line 168: Rephrase. What do you mean?
  3. Line 169: Rephrase using a verb.
  4. Line 170: Rephrase.
  5. Line 192-196: Rephrase. “Siddiqi ………. onset.
  6. Lines 214-216: Rephrase.
  7. Lines 222-224: Rephrase.
  8. Lines 327-332: Rephrase.
  9. Lines 347-352: Rephrase.

We rephrased all the paragraphs and sentences as suggested by the reviewer. Long paragraphs have been shortened and re-written in order to make sense and give a clear message.

  1. Use the above corrected subtitles (where appropriate) in the rest of the document.

We modified the questions according to reviewer’s suggestion and made the appropriate alterations to subtitles as well. 

  1. When were references 10 (lines 407-408) and 64(lines 548-549) accessed?

The years of access have been added in the reference session.

  1. In table 1 replace all the abbreviations used in the table (many of which are miss-spelt) to gain space and homogeneity.
  2. In table 1 organize the enlisted articles according to: (a) which steroid was used. For example, all Dexa reports together etc. (b) Trial design, thereafter. For example, all Retrospective trials together etc.

We reformed Table 1 in order to be clear, according to the reviewer’s suggestions. We used abbreviations and re-arranged the studies according to the type of corticosteroid used and the type of the study design.

Reviewer 3 Report

I've read the manuscript with great interest. The authors carefully describe the benefits and risks of steroids use. In my opinion adrenal insufficiency and slow dose steroid tappering should be also mentioned in the review with corresponding citations.

Some minor comments:

  1. line 24 in my opnion should be: "....(ARDS) with subsequent..."
  2. first paragraph of the introduction is identical with abstract - it would be nice to change it for the text being more interesting for the reader
  3. line 64 start the sentence with The RECOVERY trial...
  4. line 93: should be:"... had the highest benefit...."
  5. line 207: the dosing 1 mg/kg/day refers to which of the glucocorticoids? - please explain
  6. line 295: should be: "...reports of serious adverse reactions..."

Overall I find the paper very informative and interesting.

Author Response

Reviewer 3

We would like to thank the reviewer for the careful reading of our manuscript and the valuable suggestions for improving the meaning in several parts of the paper.

  1. Line 24 in my opnion should be: "....(ARDS) with subsequent..."

We corrected this phrase according the reviewer’s suggestion.

  1. first paragraph of the introduction is identical with abstract - it would be nice to change it for the text being more interesting for the reader

We would like to thank you for this observation. We modified the abstract in order to be different from the text.

  1. line 64 start the sentence with The RECOVERY trial...

           The correction has been done according the reviewer’s suggestion.

  1. line 93: should be:"... had the highest benefit...."
  2. The correction has been done according the reviewer’s suggestion.

  1. line 207: the dosing 1 mg/kg/day refers to which of the glucocorticoids? - please explain

We reviewed the original paper and we realised that the authors they refer to glucocorticoids as a general description, without determining their type.

  1. line 295: should be: "...reports of serious adverse reactions..."

We corrected the phrase according to the suggestion.

Round 2

Reviewer 2 Report

Dear Authors,

Your manuscript is accepted in present form.